

# Analysis of Systematic Biases in Tropospheric Hydrostatic Delay Models and Construction of Correction Model

Haopeng Fan[1], Siran Li[2], Zhongmiao Sun[3], Guorui Xiao[1], Xinxing Li[1], Xiaogang Liu[3]

[1]PLA Strategic Support Force Information Engineering University, Zhengzhou, 450001, China.
[2]Henan Economics and Trade Vocational College, Zhengzhou, 450046, China.
[3]Xi'an Research Institute of Surveying and Mapping, Xi'an, 710054, China.

*Correspondence to*: Haopeng Fan (fanhaopeng2008@163.com)

**Abstract.** In the fields of space geodetic techniques, such as Global Navigation Satellite System (GNSS), tropospheric zenith hydrostatic delay (ZHD) is chosen as the a priori value of tropospheric total delay. Therefore, the inaccuracy of ZHD

will definitely infect parameters like the wet delay and the horizontal gradient of tropospheric delay, accompanied by an indirect influence on the accuracy of geodetic parameters, if not dealt with well at low elevation angles. In fact, however, the most widely used ZHD model currently seems to contain millimetre-level biases from the precise integral method. We explored the bias of traditional ZHD models and analysed the characteristics in different aspects on a global annual scale. It was found that biases differ significantly with season and geographical location, and the difference between the maximum

and minimum values exceed 30 mm, which should be fully considered in the field of high-precision measurement. Then, we constructed a global grid correction model, which is named ZHD_crct, based on the meteorological data of year 2020 from ECMWF (European Centre for Medium-Range Weather Forecasts), and it turned out that the bias of traditional model in the current year could be reduced by ~50% when the ZHD_crct was added. When we verified the effect of ZHD_crct on the biases in the next year, it worked almost the same as the former year. The mean absolute biases (MABs) of ZHD will be

narrowed within ~0.5 mm for most regions, and the STD (standard deviation) will be within ~0.7 mm. This improvement will be helpful for researches on meteorological phenomena as well.

## 1 Introduction

Propagation delay caused by troposphere is inevitable in space observing technologies which employ electromagnetic waves as the primary means of detection, such as Global Navigation Satellite System (GNSS), Very Long Baseline Interferometry

(VLBI), Satellite Laser Ranging (SLR), Interferometric Synthetic Aperture Radar (InSAR), and other technologies (Bevis et al., 1994; Boehm et al., 2006; Tang et al., 2017; Drożdżewski and Sośnica, 2021; Xiao et al., 2021). Affected by weather conditions, such as air temperature, pressure, and water vapor content, this kind of delay varies strongly over time, geographical location and transmission path, and cannot be eliminated by multi-frequency combined observing, which has become one of the main bottlenecks restricting the accuracy of space geodetic surveying (Mendes, 1999; Alizadeh et al.,

2013; Younes and Afify, 2014; Yao et al., 2018; Ma et al., 2022).



To be facilitated, the tropospheric delay is usually divided into hydrostatic and non-hydrostatic (wet) components (Davis et al., 1985). As the main influence, the hydrostatic component accounts for ~90% of the total delay, which reaches 20 m or even larger at low range of elevation angle and still remains ~2 m in the zenith direction (Niell, 1996; Chen and Herring, 1997; Penna et al., 2001). In data processing of technologies such as GNSS and VLBI, zenith hydrostatic delay (ZHD) is

usually taken as the a priori value of the total zenith delay, and the estimated term as the zenith wet delay (ZWD); finally, both of them are respectively multiplied by the corresponding mapping function and then summed up to obtain a slant-path one (Mendes et al., 2002). To some extent, ZHD seems to be the key to determining the total delay.

Fortunately, ZHD models, such as Hopfield (1971), Saastamoinen (1972), are capable of approximating the actual situation well. Moreover, with the help of succeeding researchers, uncertainties of model ZHDs have been limited in sub-millimetre

level, especially using closed formulae induced from precise integral method based on hydrostatic equilibrium condition (Davis et al., 1985; Zhang et al., 2016). Thus, in fields including GNSS and VLBI, calculated figures from the ZHD models are widely used even as true values (Wang et al., 2005;Tuka and El-Mowafy, 2013; Liu et al., 2017; Feng et al., 2020), which facilitate quite a few advanced models, such as GPT series, VMF series (Boehm et al., 2006; Boehm et al., 2007; Böhm et al., 2015; Landskron and Böhm, 2018) and extrapolation models (Li et al., 2018; Hu and Yao, 2019; Li et al., 2020;

Wang et al., 2022).

However, hydrostatic equilibrium will be broken if vertical wind acceleration occurs, which shall probably be influential to the accuracy of traditional ZHD models depicted by closed formulae. In fact, it has been noticed that these models show some certain systematic biases when compared with the precise integral method, which vary with locations or time yet are easily neglected (Liu et al., 2000; Chen et al., 2009; Yan et al., 2011; Dai and Zhao, 2013; Zhang et al., 2016; Feng et al.,

2020). In the direction of zenith or high-altitude angle, the estimates of technologies like GNSS can hardly be affected by this kind of bias. That's because the hydrostatic mapping function is nearly equal to the wet one, and the ZHD and ZWD biases rightly offset each other. Nevertheless, when it comes to low elevation angles, the difference between mapping functions of the two components will lead to slant-path delay errors up to ~10 mm (Fan et al., 2019) and coordinate errors of ~2 mm when mapped on to the vertical direction by the rule of thumb (Boehm et al., 2006), which furtherly affects the

accuracy of geodetic solutions.

In addition, biases of ZHD will be transferred to ZWD, which is often pursued by ZTD minus ZHD. Since ZWD is closely related to precipitable water vapor (PWV), roughly 0.15–0.25 times of ZWD (Yao et al., 2016), ZHD biases will cause PWV errors indirectly, which is undoubtedly unfavourable for studying meteorological phenomena like the atmospheric water cycle, and forecasting disastrous weather, including rainstorms and typhoons.

From the point of view of situations above, we analysed the characteristics of model ZHD biases from different aspects, and constructed a set of grid correction model to provide reliable reference for improving various solutions to space geodetic surveying data and obtaining precise meteorological parameters such as PWV.



## 2 Data Preparation

### 2.1 Profile meteorological data

The profile meteorological data were needed to calculate ZHD reference values based on the integral method, i.e., ray tracing. Data observed by radiosondes or meteorological products provided by international authorities are all suitable to achieve reference values. Our ZHD reference values were obtained mainly based on the multilayer grid meteorological data provided by the fifth generation ECMWF reanalysis (ERA5) dataset, which have been validated well in tropospheric delay calculations (Abdelfatah et al., 2015; Bekaert et al., 2015; Graham et al., 2019; Sun et al., 2019; Dogan and Erdogan, 2022).

The ERA5 profile data are mostly valid at about 30–40 km high. As there still exists a small amount of air above, when the valid height of data was exceeded during experiments, they were complemented until 86 km using the American standard atmospheric model COESA 1976 (Minzner, 1977).

Given the low vertical resolution of ERA5 and COESA 1976, Equation (1)–(3) were used to interpolate air pressure, temperature and humidity between adjacent layers (Nafisi et al., 2012), where the interpolating step follows the criterion

described by Rocken et al. (2001).

$$p_{\text{int}} = p_i \exp[-\frac{(h_{\text{int}} - h_i)g_i}{R_d T_{v,i}}] \tag{1}$$

$$T_{\text{int}} = T_{i+1} + \frac{T_i - T_{i+1}}{h_i - h_{i+1}}(h_{\text{int}} - h_{i+1}) \tag{2}$$

$$p_{w,\text{int}} = \begin{cases} p_{w,i} \exp[\frac{h_{\text{int}} - h_i}{h_{i+1} - h_i}\log(p_{w,i+1}/p_{w,i})], & \text{if } p_{w,i} > 0 \\ \frac{p_{w,i+1}}{h_{i+1} - h_i}(h_{\text{int}} - h_i), & \text{if } p_{w,i} = 0 \end{cases} \tag{3}$$

The subscript $i$ in Equation (1) – (3) refers to the meteorological elements (temperature $T$, total air pressure $p$ and the water

vapour pressure $p_w$); $p_{int}$, $T_{int}$ and $p_{w,int}$ represent the air pressure, temperature and water vapour pressure at the interpolation height $h_{int}$, which locates between the $i^{\text{th}}$ and $(i+1)^{\text{th}}$ layer; $T_{v,i}$ means the virtual temperature and its calculation follows Equation (4) (Hofmeister, 2016).

$$T_{v,i} = \frac{T_i p_i}{p_i - 0.378 p_{w,i}} \tag{4}$$

### 2.2 Calculation of ZHD reference data

According to the principle of ray tracing (Thayer, 1967; Nafisi et al., 2012; Eriksson et al., 2014), ZHD can be accurately achieved by taking the integrals of refractive parameters in the sky above the study site, specifically as seen in Equation (5).



$$\text{ZHD}_{\text{reference}} = \int_{h0}^{H} [n^H(h)-1]\mathrm{d}h \approx \sum_{i=1}^{k-1}(n_i^H-1)\Delta h_i = 10^{-6}\sum_{i=1}^{k-1}N_i^H\Delta h_i \tag{5}$$

Where $n^H(h)$, $h_0$, $H$ and $k$ represent hydrostatic delay, the refractive index of hydrostatic delay at height $h$, geodetic height of the site, maximum tropospheric thickness, and the total number of divided tropospheric layers, respectively. $n_i^H$ and $N_i^H$ are the hydrostatic component refractivity and refractive index respectively, and $\Delta h_i$ denotes the thickness between subdivide layers. Therein, $N_i^H$ can be calculated through Equation (6) (Davis et al., 1985).

$$N_i^H = k_1\frac{p_i}{T_i} \tag{6}$$

Where $k_1$ is a constant (77.6890 K/hPa) selected from the "best average" parameters of Rüeger (2002), while $p_i$ and $T_i$ represent the hydrostatic component air pressure and air temperature between the corresponding height layers, respectively. Given accurate meteorological parameters, the integral results shall naturally be of high reliability, thus serving as reference or actual values in atmospheric delay effect analysis (Hobiger et al., 2008; Hofmeister and Böhm, 2017; Osah et al., 2021). Since ECMWF possesses high-quality meteorological parameter values of profile grids at both global and yearly scales, ZHD calculated using these data were treated as the reference value in subsequent statistical analysis and modelling.

**2.3 Traditional ZHD model**

According to findings of pioneers (Saastamoinen, 1972; Davis et al., 1985; Penna et al., 2001; Tregoning and Herring, 2006; Leandro et al., 2008), ZHD values can be determined by closed formula as Equation (7)

$$\text{ZHD}_{\text{Empirical}}(p_0,h_0,\varphi) = C\times\frac{p_0}{1-0.0026\cos(2\varphi)-2.8\times10^{-7}h_0} \tag{7}$$

Where $p_0$, $h_0$ and $\varphi$ denote the air pressure (mbar), height (m) and geodetic latitude (°) of the study site, while the constant $C$ in most cases is set as 0.0022768, refined by Davis et al. (1985); or it could be set as 0.0022794, modified by Zhang et al. (2016), which is claimed to be able to cut bias down within 1 mm. Two values of $C$ were both studied in the following part. Obviously, traditional models are simpler and more practical than the integral method since only the pressure and position parameters of the site are required. In the experiments, we exploited the ECMWF surface data for meteorological inputs of traditional models.

**3 Biases' Characteristics and the Correction Model**

**3.1 Location-specific analysis**

Based on the data prepared in Section 2, we obtained the ZHD model biases by subtracting model ZHDs from reference ZHDs. Twelve sites, divided into 4 groups, were chosen to learn the time-varying characteristics, whose locations and climate types are depicted in Figure 1 and Table 1. Three-year (2019–2021) time series of biases of study sites were drawn according to Equation (5) and (7), and two types of constant $C$ were used (Figure 2). Here we note Equation (7) as model



SAAS$_D$ for $C = 0.0022768$ and model SAAS$_Z$ for $C = 0.0022794$. Additionally, the ZHD mean absolute biases (MAB) and standard deviation (STD) of biases for the two traditional models in each study site are displayed in Table 2. The MAB is calculated as $\frac{1}{N}\sum_{i=1}^{N}|\text{Bias}_i|$, where $\text{Bias}_i$ means the bias of the $i^{\text{th}}$ grid point, and $N$ means the total number of grid points.

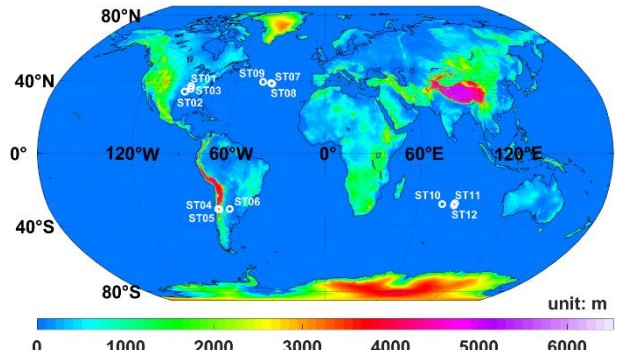

**Figure 1: 10 arc-minute global land topography. White circles are the locations of 12 sites, and colour bar represents global**
**surface fluctuations. Topography data were abstracted from ETOPO1 Ice Surface grid model. The colour bar denotes the altitude.**

**Table 1**
*Geodetic Coordinates of Study Areas and Corresponding Climate Types*

| Group | Site abbreviation | Latitude (Unit: °) | Longitude (Unit: °) | Altitude (Unit: m) | Climate type | Climate name |
|---|---|---|---|---|---|---|
| | ST01 | 36.83 | -89.17 | 94.9 | Cfa | Humid subtropical climate |
| 1 | ST02 | 33.80 | -92.30 | 60.5 | Cfa | Humid subtropical climate |
| | ST03 | 35.33 | -88.83 | 136.1 | Cfa | Humid subtropical climate |
| | ST04 | -30.50 | -69.50 | 3094.0 | BWk | Cold desert climate |
| 2 | ST05 | -30.67 | -68.67 | 1014.3 | BWh | Hot desert climate |
| | ST06 | -30.50 | -62.00 | 94.0 | Cfa | Humid subtropical climate |
| | ST07 | 38.50 | -36.33 | 0 | — | — |
| 3 | ST08 | 38.17 | -35.50 | 0 | — | — |
| | ST09 | 39.17 | -41.83 | 0 | — | — |
| | ST10 | -27.83 | 75.17 | 0 | — | — |
| 4 | ST11 | -27.33 | 83.33 | 0 | — | — |
| | ST12 | -28.50 | 82.83 | 0 | — | — |

Note: Climate type of each site complies with Köppen-Geiger climate classification, and for more details please refer to Peel et al. (2007) and Beck et al. (2018).







**Figure 2: Bias time series of model SAAS$_D$ and SAAS$_Z$. (a)–(l) represent the situations of the 12 sites respectively. Light red dot line and light blue dot line show the mean value of bias series from SAAS$_D$ and SAAS$_Z$ respectively.**

**Table 2**
*Mean Absolute Bias (MAB)/Standard Deviation (STD) of SAAS$_D$ and SAAS$_Z$ Biases (Unit: mm)*

| Site code | SAAS$_D$ | SAAS$_Z$ |
|---|---|---|
| ST01 | 2.98/0.43 | **0.45/0.42** |
| ST02 | 2.98/0.45 | **0.46/0.45** |
| ST03 | 2.93/0.43 | **0.44/0.43** |
| ST04 | **0.92/0.97** | 2.29/0.98 |
| ST05 | **1.18/1.46** | 2.55/1.46 |
| ST06 | 2.92/0.52 | **0.47/0.51** |
| ST07 | 2.93/0.38 | **0.37/0.38** |
| ST08 | 2.92/0.37 | **0.36/0.37** |
| ST09 | 2.91/**0.38** | **0.36**/0.39 |





| | | |
|---|---|---|
| ST10 | 2.80/0.33 | **0.28/0.33** |
| ST11 | 2.82/**0.34** | **0.30**/0.35 |
| ST12 | 2.82/0.34 | **0.29/0.34** |

Note: Bold numbers represent minimum absolute values in row comparisons.

It can be seen from Figure 2 that if no correction applied, $SAAS_D$ model would generate about 2.5-millimetre biases; by contrast, those from $SAAS_Z$ seem to be restricted within 0.5 mm mostly (10 sites in 12 in Table 2). Yet, in fact, there exists unknown high-frequency residual information in both two ZHD models, which might be speculated that it should be related to the vertical pressure gradient in each region. Also, it turns out that $SAAS_Z$ only increase by ~ 2 mm overall, with the periodic trend still reserved, and the biases' STD of $SAAS_Z$ keeps almost the same with $SAAS_Z$ (Table 2). Besides, from

Figure 2, the STDs in marine areas are generally smaller than those on land, and the value of STD seems to be related to the complexity of climatic condition; for example, the sites in Group 1 show little variety in STDs, while those in Group 2 experience a quite different thing.

We took $SAAS_D$ for example in the following part to analyse the characteristics in further. The correlation coefficient matrix of biases of 12 sites was shown in Figure 3, as well as their spectrum analysis diagram (Figure 4) and percentage of annual

and half annual items in total energy (Figure 5).

It can be inferred from Figure 3 that the correlation coefficient gets larger if the two sites are closer. For instance, the coefficients of the sites within the Group are apparently lager than those between Groups; also, as ST02 is away from the other two in Group 1, the correlation appears weaker. The same phenomenon can be observed from the relationship between ST06 and the other two in Group 2. When we compare the correlation between Group 1 and Group 2, it can be seen that the

correlation is also generally larger in areas with the same climate category or at similar elevation. When it comes to the sites in Group 3 and Group 4, it turns out that the biases in marine areas also varies with the location, and the correlation between them seems stronger than those between the sites with the same distance spacing on land. Combined with the spectrum diagram (Figure 4), it can be seen that the biases for most sites are generally influenced by annual and semi-annual signals, and the energy intensity varies significantly with site's location: Regions with high altitude appear more sensitive to seasons,

such as ST04 and ST05, while situations for those in the ocean in south hemisphere are opposite, such as ST10–ST12 (Figure 5).



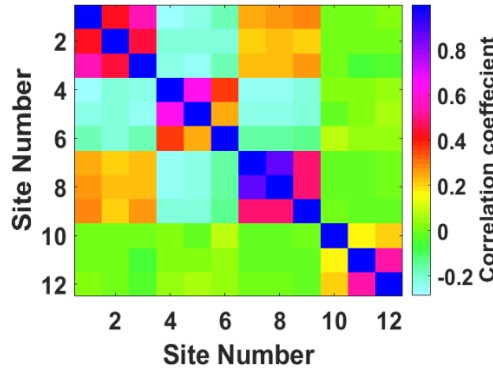

**Figure 3: Correlation coefficients of bias time series of the 12 sites.**

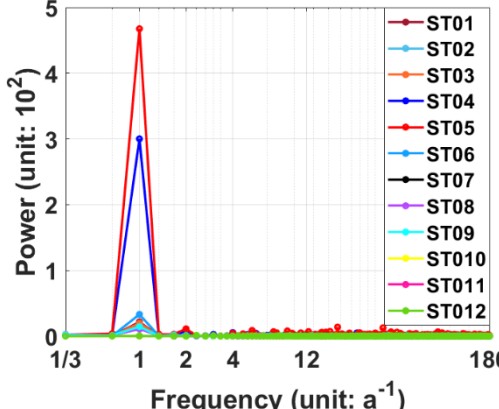


**Figure 4: Temporal frequency spectrograms for ZHD biases of model SAAS$_D$.**

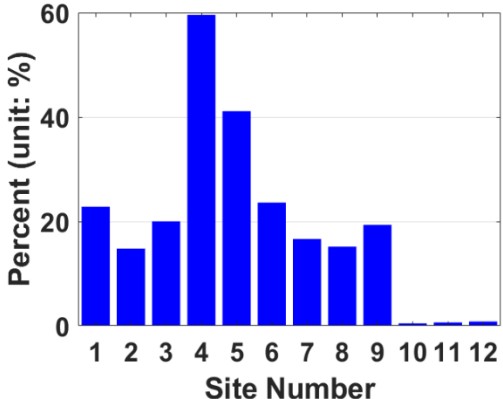

**Figure 5: Percentage of annual and semi-annual items in total energy for ZHD biases of model SAAS$_D$.**



## 3.2 Global analysis

To further analyse the global distribution characteristics of ZHD systematic biases, we calculated the 1 arc-degree grid values of ZHD biases during January–December, 2020, and depicted their monthly averaged distributions (Figure 6) with statistics listed in Table 3. Herein, we have some conclusions.

1) It confirms the seasonal characteristics, which is reflected in varying degrees throughout the world. For instance, the bias values in northern hemisphere winter are averagely negative, while those in the southern hemisphere, especially in high-

altitude areas, are positive. Then, with season shifting, the values in the northern hemisphere gradually become positive and those in the southern hemisphere winter go to the opposite way. This annual regression phenomenon can also be found by combining the MABs and STDs of the global monthly averaged biases in Table 3.

2) Combining with Figure 1, we find that the biases have a different sign in regions with different altitude. In the northern hemisphere in winter, there are more positive values in high-altitude areas (such as the Qinghai Tibet Plateau in Asia and the

Andes Mountains in America), yet more negative ones in marine areas (such as the North Atlantic and the East Pacific) or low-altitude land (such as northwest of Eurasian continent and northeast of North America). However, this pattern is opposite in Antarctica and areas with high altitude on Greenland. The preliminary judgment may be subjected to the fact that those regions are in the polar high-pressure zone all year round, and the vertical pressure gradient force is usually less than the atmospheric gravity; at this time, the hydrostatic equilibrium is broken, and the closed formula deduced from this

condition deviates. Looking back at $SAAS_Z$, if this model is used, biases in areas with high altitude will probably become worse.

3) By and large, biases in the southern hemisphere (except Antarctica) varies with latitude even in different seasons. Considering that the ocean area in the southern hemisphere is much larger than the land, and the meteorological condition is dominated by oceans, which is latitudinally dependent, this probably leads to the latitudinal distribution. As a

counterexample, that pattern is weak in the northern hemisphere.

4) Overall, the MAB of $SAAS_D$ over the whole year is ~0.77 mm, not that large, but it changes significantly with season and geographical location, and the difference between the maximum and minimum values exceed 30 mm, especially in the northern hemisphere in summer (Table 3). Hence, such a kind biases should be fully considered in the field of high-precision measurement.

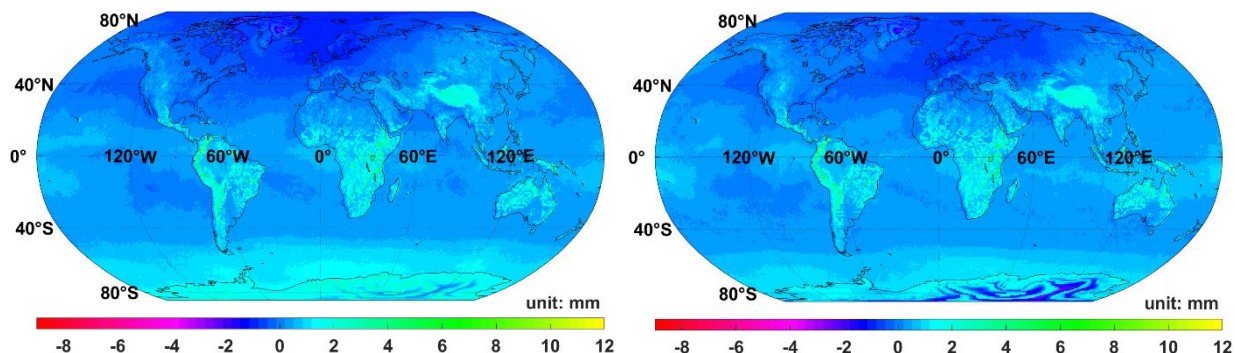




(a) January

(b) February

(c) March

(d) April


(e) May

(f) June

(g) July

(h) August





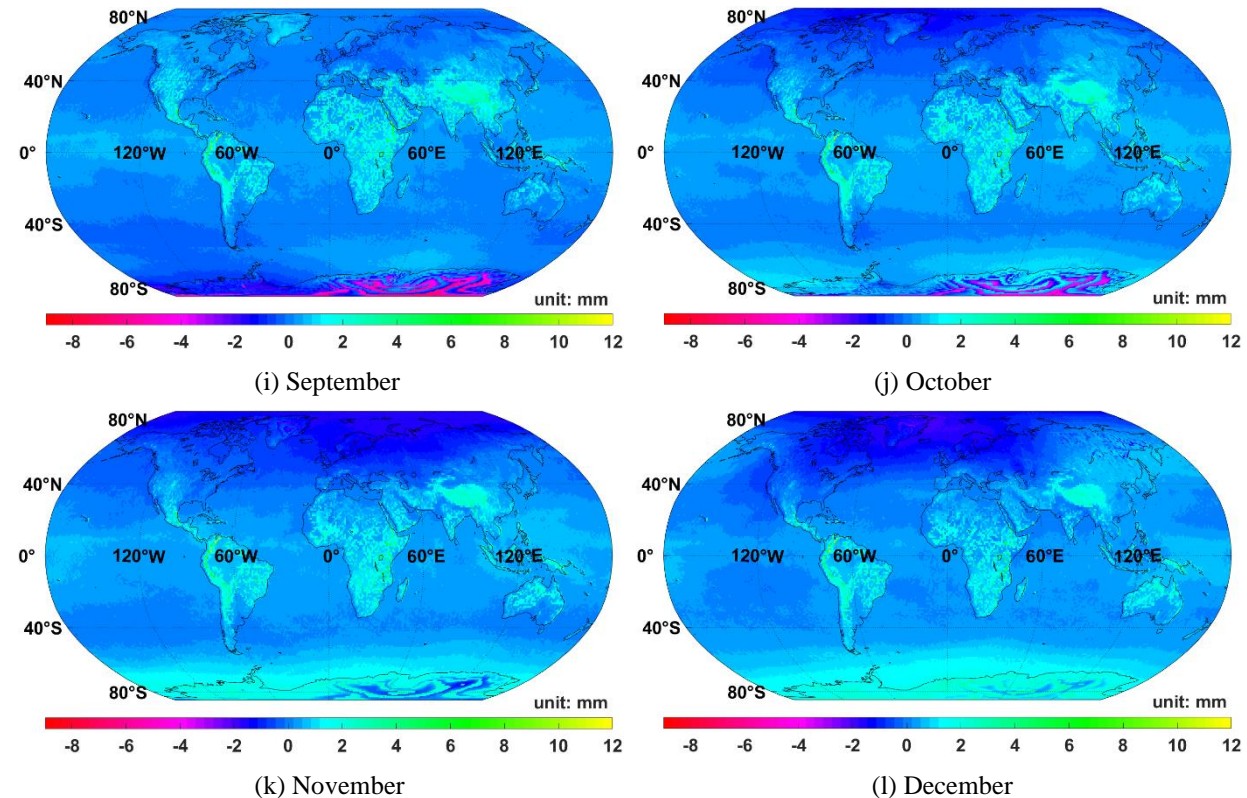

(i) September           (j) October

(k) November           (l) December

**Figure 6: Global distribution maps of monthly averaged ZHD systematic biases between SAAS$_D$ ZHD and integral ZHD. (a)–(l) represent the monthly averaged biases from January to December, 2020. In order to distinguish the variations between different months, we use the same scale colour bars in pictures.**

**Table 3**

*Statistics of Systematic Biases in Different Months and the Whole Year (Unit: mm)*

| Month | Min bias | Max bias | MAB | STD |
|---|---|---|---|---|
| January | -10.92 | **13.64** | 0.72 | 0.87 |
| February | -7.76 | 13.60 | 0.64 | 0.79 |
| March | -11.37 | 13.25 | 0.59 | 0.90 |
| April | -12.11 | 13.23 | 0.68 | 1.09 |
| May | -13.06 | 12.57 | 0.92 | 1.41 |
| June | -14.10 | 12.25 | **1.03** | 1.56 |
| July | -14.42 | 11.50 | 0.97 | **1.57** |
| August | **-18.00** | 12.09 | 0.80 | 1.47 |





| | | | | |
|---|---|---|---|---|
| September | -11.91 | 12.10 | 0.62 | 1.13 |
| October | -10.24 | 12.79 | 0.65 | 0.97 |
| November | -7.42 | 12.47 | 0.79 | 1.00 |
| December | -8.90 | 12.65 | 0.82 | 1.03 |
| The whole year | -18.00 | 13.64 | 0.77 | 1.20 |

Note: Bold numbers represent minimum absolute values in column comparisons.

### 3.3 Grid correction model

Based on the periodic characteristics, the biases were approximated using a trigonometric function as Equation (8).

$$\delta_{ZHD} = a_0 + a_1 \cos(2\pi t / 365.25) + a_2 \sin(2\pi t / 365.25)$$
$$+ a_3 \cos(4\pi t / 365.25) + a_4 \sin(4\pi t / 365.25)$$

(8)

Where $\delta_{ZHD}$ is the approximated ZHD bias, $a_0 - a_4$ represent annual average, annual and semi-annual periodic term coefficients, respectively, and $t$ stands for the day of year (DOY). The parameters were estimated using the biases of each grid point on a global scale in 2020, and the distribution of $a_0$, $\sqrt{a_1{}^2 + a_2{}^2}$ and $\sqrt{a_3{}^2 + a_4{}^2}$ were depicted by Figure 7. Because of the location-related feature (based on Section 3.1), a grid correction model plus space interpolation shall be applicable to recover the bias at any position at any time during year 2020. Five model parameters ($a_0 - a_4$) will be needed for each grid point. Once the $\delta_{ZHD}$ is calculated, it will just serve as the correction of ZHD model. We call it ZHD_crct.

From Figure 7 we have conclusions. 1) In mid & low-latitude areas, annual average of ZHD biases clearly varies with terrain (or air pressure, since pressure generally decrease with height increasing), which is maintained between -0.2–0.4 mm in ocean and plain areas, and ranges from 1.0 to 10.0 mm in high-altitude areas like plateaus and mountains. Results seem to be reversed in cold high-latitude areas, such as Greenland (Arctic) and some of Antarctica, for annual average decreases to -4 mm or even lower.

2) The annual amplitudes in the mid & low-latitude marine areas are significantly smaller than those in high-latitude land areas. Therefore, if we don't ask for high accuracy, the annual variation of these areas can be directly neglected. However, on the land and in high-latitude areas, especially in Antarctica, Greenland, southern and Western Asia, northeast and northwest Africa and other regions, this variation should be well considered, for the annual impact reaches ± 4mm or more.

3) The amplitudes of semi-annual periodic term are insignificant, by and large, yet in areas with strong annual signal, the semi-annual effect is still obvious enough, even reaching ~1.6 mm. In addition, in the ocean areas near the equator, the semi-annual signal is outstanding from surroundings, which remains further study.



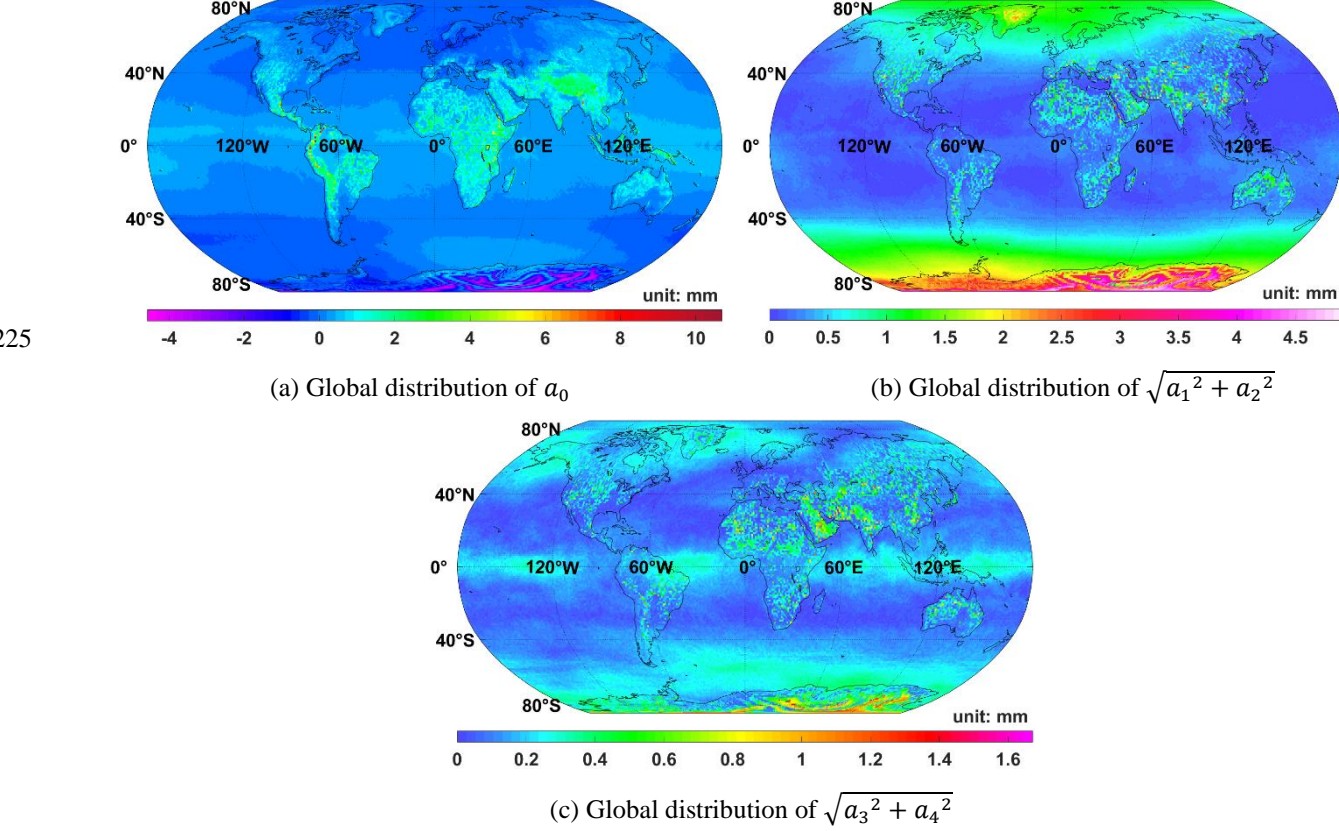

(a) Global distribution of $a_0$        (b) Global distribution of $\sqrt{a_1^2 + a_2^2}$

(c) Global distribution of $\sqrt{a_3^2 + a_4^2}$

**Figure 7: Global distribution of coefficients in periodic function. Changes in the constant term, annual and semi-annual amplitudes are described in (a)–(c), respectively.**

## 4 Validation

### 4.1 Internal coincidence examination

We collected the residual biases of ZHD during 2020 after ZHD_crct being added for checking the model's internal coincidence. As the global STD in summer in the northern hemisphere reaches the maximum, here the monthly averaged biases of July were painted out for instance by Figure 8a, as well as the distribution chart of their frequency number before and after correction by Figure 9a. Also, the annual averaged MABs and their statistical feature can be referred to Figure 8b and Figure 9b. Comparing Figure 8 and Figure 6g, it can be seen that the corrected biases are less than ±0.5 mm in most regions of the world, even in mountainous and plateau regions; positive and negative values are nearly evenly distributed in the northern and southern hemispheres; and from Figure 9, the MABs and STDs are sharply reduced by ~50% both over July and the whole year. However, biases larger than ± 2mm still exist in Antarctica, Andes Mountains and parts of Asia-Africa continent, and the improvement near equator appears worse than other places at mid & low latitude, keeping similar to the situation of semi-annual amplitudes (Figure 8b).



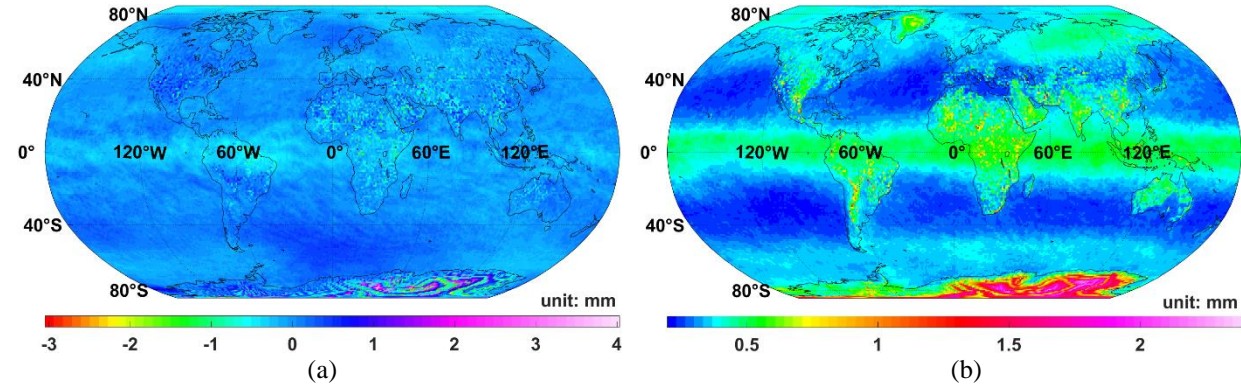

**Figure 8: MAB over July (a) and the whole year (b) of 2020 after correction by grid model.**

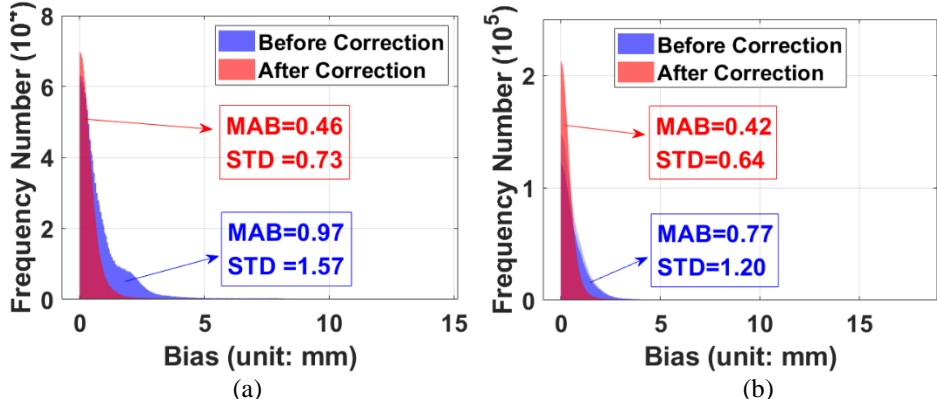

**Figure 9: Frequency distribution of MABs over July (a) and the whole year (b) of 2020 before and after correction.**

250 Figure 10 shows the MAB and STD of biases in different dimensions before and after correction, in which (a) represents the global MABs and STDs on different DOYs, (b) represents the MABs and STDs at different latitudes, (c) represents the MABs and STDs in regions with different air pressures, (d) represents the MABs and STDs in regions with different temperatures, and (e) represents the MABs and STDs in regions with different heights.

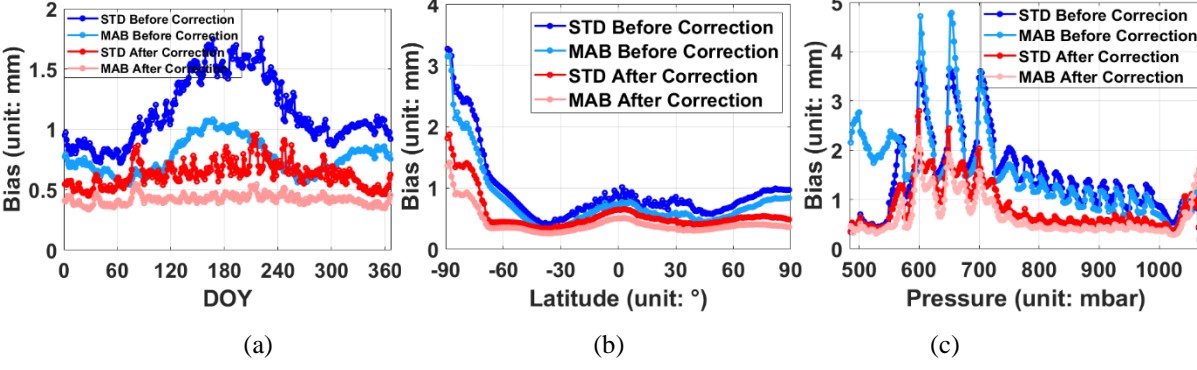

255              (a)                      (b)                    (c)



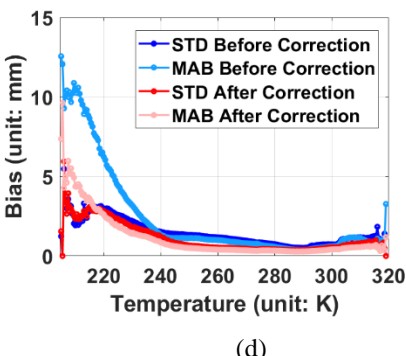
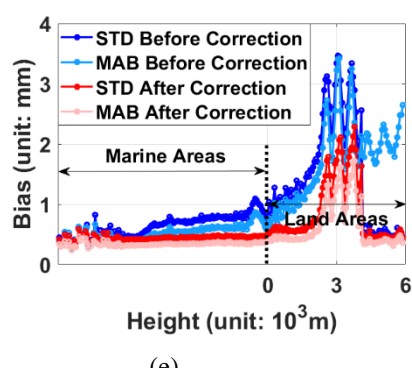

(d)                                    (e)

**Figure 10: The MAB and STD of biases in different dimensions before and after correction over 2020. (a) represents the global MABs and STDs on different DOYs, (b) represents the MABs and STDs at different latitudes, (c) represents the MABs and STDs in regions with different air pressures, (d) represents the MABs and STDs in regions with different temperatures, and (e) represents the MABs and STDs in regions with different heights.**

As can be seen from Figure 10, the corrected biases have been significantly cut down in different dimensions. Figure 10a shows that the annual and semi-annual trends of biases are basically removed. The biases are weakened the most in winter and summer, and the STDs decrease by about 50% in all periods of the year. Figure 10b shows that the improvement in high-latitude regions and those in southern hemisphere with mid & low latitudes is superior; by contrast, the reduction of STDs of biases in the Antarctic region doesn't perform well enough compared with that in others. Figure 10c shows that the reduction in the low-pressure (600–750 mbar) areas is weaker than that in other areas; these areas are mainly near the south pole, which is basically consistent with the previous conclusion. At the same time, the improvement in the high-pressure (above 1000 mbar) areas is also insignificant, which needs to be further studied. Figure 10d shows that in the low temperature (below 240K) areas (also mainly near the south pole), the improvement doesn't outstand others; although the MABs are reduced by half, the STDs are not significantly improved compared with that before correction. Figure 10e shows that the improvement is more obvious in land areas with an altitude below 3000 m or over 5000 m.

In terms of the views above, it can be summarized that if no correction is made near the south pole, the error of ZHD by SAAS$_D$ can be as large as 10 mm or worse; in the northern hemisphere in summer, in the regions with pressure lower than 750 mbar, or with the temperature lower than 260 K, or most land areas, if no correction is applied, the biases of ZHD model will be larger than 1 mm generally.

### 4.2 External coincidence examination

In order to learn the external coincidence of this grid model, we collected the global biases over the year 2021 before and after correction using ZHD_crct, which was fed with coefficients solved according to 2020 bias series. Results of four months' MAB were painted out for comparison (Figure 11), also the statistics of biases (Table 4) and the MAB/STD of biases in different dimensions (Figure 12) were exhibited.





(a) MABs in January 2021 before correction    (b) MABs in January 2021 after correction

(c) MABs in April 2021 before correction    (d) MABs in April 2021 after correction

(e) MABs in July 2021 before correction    (f) MABs in July 2021 after correction



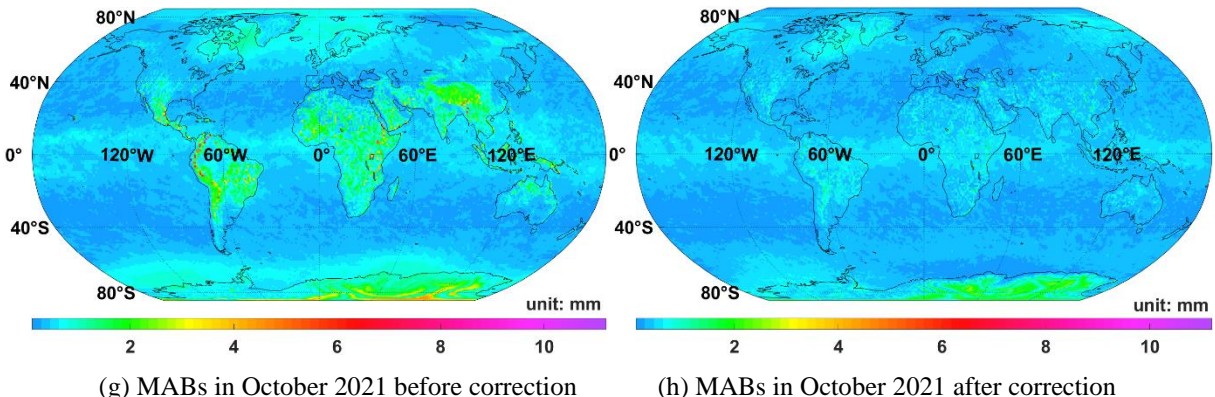

(g) MABs in October 2021 before correction   (h) MABs in October 2021 after correction

**Figure 11: Global MABs before (left ones) and after (right ones) correction by grid model over January (a–b)/ April (c–d)/ July(e–f)/ October(g–h) in 2021. Please note that the colour bar of each month is different.**

**Table 4**

*Statistics of Systematic Biases Before/After Correction in Different Months and the Whole Year of 2021(Unit: mm)*

| Month | Min bias | Max bias | MAB | STD |
|---|---|---|---|---|
| January | -9.28/**-7.44** | 13.27/**8.53** | 0.70/**0.47** | 0.84/**0.65** |
| February | -9.43/**-8.01** | 13.88/**8.03** | 0.77/**0.50** | 1.00/**0.68** |
| March | -9.70/**-6.39** | 13.27/**8.57** | 0.59/**0.44** | 0.89/**0.63** |
| April | -14.71/**-11.73** | 12.84/**8.95** | 0.68/**0.45** | 1.15/**0.69** |
| May | -14.14/**-9.64** | 12.73/**10.57** | 0.93/**0.45** | 1.42/**0.71** |
| June | -16.90/**-9.27** | 12.17/**10.15** | 1.11/**0.44** | 1.73/**0.70** |
| July | -13.75/**-8.99** | 11.93/**10.45** | 1.03/**0.43** | 1.67/**0.68** |
| August | -16.59/**-7.96** | 12.38/**9.63** | 0.84/**0.45** | 1.54/**0.72** |
| September | -12.17/**-8.16** | 11.58/**8.91** | 0.63/**0.43** | 1.18/**0.71** |
| October | -9.53/**-8.56** | 12.84/**7.35** | 0.60/**0.43** | 0.93/**0.64** |
| November | **-6.16**/-8.56 | 12.93/**5.11** | 0.74/**0.38** | 0.97/**0.52** |
| December | **-8.63**/-9.05 | 13.87/**7.08** | 0.86/**0.40** | 1.09/**0.53** |
| The whole year | -16.90/**-11.73** | 13.88/**10.57** | 0.79/**0.44** | 1.25/**0.66** |

Note: Bold numbers represent minimum absolute values in row comparisons.




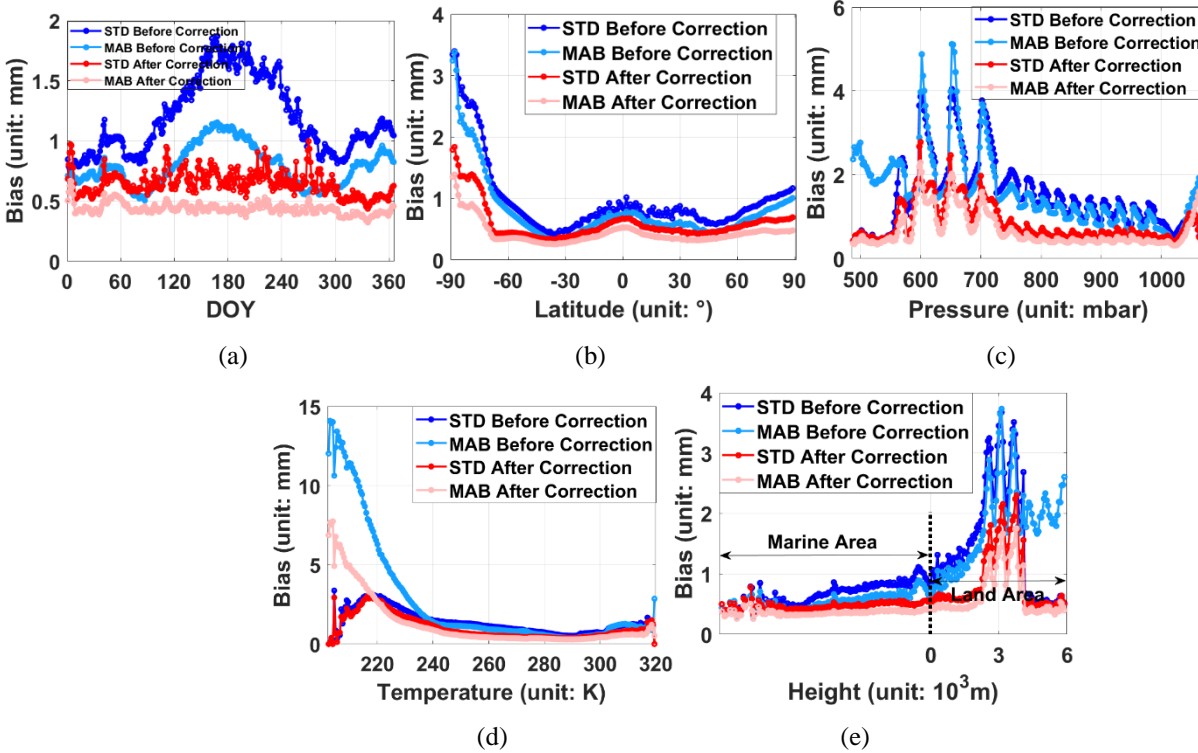


(a)         (b)         (c)

(d)         (e)

**Figure 12: The MAB and STD of biases in different dimensions before and after correction over 2021. (a) represents the global MAB and STD on different DOYs, (b) represents the MAB and STD at different latitudes, (c) represents the MAB and STD in regions with different air pressures, (d) represents the MAB and STD in regions with different temperatures, and (e) represents the MAB and STD in regions with different heights.**

From the charts above, it is clear that the correction made according to the biases of 2020 still works in 2021, and the improvement over the two years seems alike, which means that it is feasible to use historical meteorological data to predict the future bias corrections.

**5 Conclusions**

We made a detailed analysis of the biases from the traditional ZHD model. If no correction applied, $SAAS_D$, one of the most
widely used models, would generate biases at millimetre level. Although the MAB over the whole year is ~0.77 mm, biases differ significantly with season and geographical location, which reach ±10 mm or even larger, and the difference between the maximum and minimum values exceed 30 mm. Therefore, it should be fully considered in the field of high-precision measurement, especially for the regions near the south pole, and those with pressure less than 750 mbar, temperature less than 260 K, and most land aeras.
In order to cut down the biases, a grid model, ZHD_crct, was constructed exploiting a periodic function based on the meteorological data of the current year from ECMWF. Generally, ZHD biases were reduced by ~50% after correction, with



annual and semi-annual terms well removed, and the model still works when used to rectify the biases in the next year, which can be inferred that this model may be possible to forecast corrections in quite a few years. Yet, unfortunately, our model only shows expert in detrending the seasonal impacts, with unknown high-frequency residuals still remained to be

studied further, which is likely to be recovered by vertical velocity of air.

In addition, nowadays profile meteorological parameters can be provided from various organizations, such as MERRA2 data products from the NASA Global Modeling and Assimilation Office, NCEP reanalysis data products, the Integrated Global Radiosonde Archive (IGRA) from the National Climatic Data Center of America (NCDC), and the University of Wyoming Atmospheric Science Radiosonde Archive. Hence, ZHD bias correction models could be further developed based on

different datasets in follow-up studies.

*Code and data availability.* The current version of *ZHD_crct model* and its corresponding documents are available at DOI: *10.5281/zenodo.6948148 (Fan et al., 2022)*. ERA5 data can be obtained from Copernicus Climate Data Store (https://cds.climate.copernicus.eu/cdsapp#!/dataset/reanalysis-era5-pressure-levels?tab=form).    The    U.S.    Standard

Atmosphere 1976 model can be referred at https://www.pdas.com/atmosTable1SI.html. ETOPO elevation dataset can be downloaded at https://ngdc.noaa.gov/mgg/global/global.html.

*Author contributions.*

Haopeng Fan designed the research framework, writing the paper and conducted most of the code implementation and data

analysis. Siran Li was involved in data collation and revising of the paper. Zhongmiao Sun was responsible for supervision. The work in this paper was funded by the Programs from Guorui Xiao and Xiaogang Liu. Xinxing Li revised the paper.

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
