# Peer review of "Analysis of Systematic Biases in Tropospheric Hydrostatic Delay Models and Construction of Correction Model"

_EGUsphere, 2022_

## Author Response (AR1)

**Appendix 1.**

<h2 align="center">Comments and replies</h2>

Thanks for editorial team's work and both of the referees' suggestions and opinions. Here we present all the comments from referees and the corresponding replies for editor's further decision.

**Appendix table 1.**

| NO. of comments | NO. of referees | Comments | Responses | Changes in manuscript |
|---|---|---|---|---|
| 1 | 1 | How much is the difference between synoptic observations (like T and P) and surface data of the ECMWF model? | The surface meteorological data of ECMWF are in good agreement with the synoptic observations, which has been verified by researchers (Boccara et al., 2008; Decker et al., 2012;Tian et al., 2018 Aparecido et al., 2019; Aminreza et al., 2021). Here we presented the surface temperature and pressure results from ECMWF at UTC 0:00 on January 1st, 2020 as well as in-situ observations from more than 2,000 stations at the same time, which were taken by a broad variety of organisations, including International Civil Aviation Organisation, Air Force Weather Agency and World Meteorological Organisation. Appendix figure 1 depicts the temperature of two datasets and Appendix figure 2 for pressure. Also, we plotted the differences between the two datasets, which are | We didn't add this part in our manuscript, as the data we used were mainly from ECMWF and they were quite sufficient for our study. |

| | | | shown in Appendix figure 3.
It can be seen that there is little difference between two types of data generally. Since the in-situ data are instantaneous observations, while the ECMWF data are reanalysed results, several large biases still exist at some certain sites. | |
|---|---|---|---|---|
| 2 | | I am not sure about the way of your selection for the calculation of tradition and reference outputs. Please explain why you did not use other data sources for the traditional model, such as synoptic data. | Thanks for your suggestion. The purpose of this paper is to explore the deviation between the commonly used dry delay model and the integration method, which is supposed to be the most accurate theoretically. This kind of deviation is very easy to be ignored, but it does exist. Then, we found that it complies a certain rule in the global scope and the whole year, which is one of the main tasks of this paper.
As for the usage of synoptic data you suggest, it is also a possible way. However, if we want to study the relation between the traditional model and integration method, we need to use the measured synoptic data of vertical profiles correspondingly. At present, such data can only be obtained by means | We added a sentence to make us clear at Line 76. |

| | | | of sounding balloons or radiosonde. Although there are indeed many such stations around the world, it is still not enough for us to study the global bias characteristics, including marine regions. Therefore, we chose ECMWF data, which are more abundant. | |
|---|---|---|---|---|
| 3 | | Eq. 8: Please add a few sentences about why you have employed the Fourier function here. Is that because of other researches results, e.g., the GPT model? Please add a reference in this case. | Thank you very much for your reminder. We used the Fourier function mainly because of the periodical feature of the biases, which was exposed when we chose 12 sites over different regions and when it came to the annual performance in global analysis. Of course, this method is not firstly proposed by us, and some researches have also used it in models such as GPT and many related studies, so we really shall add references for annotation. | we added 2 references after Eq. 8 at Line 230. |
| 4 | 2 | Line 37: "To some extend..." - I suggest moving this sentence to the end/middle of the paragraph, starting at line 56. It is unclear here why the ZHD is the key to the total delay determination, and you explained it | Thanks for your suggestion, and we added an explanation before this sentence. | We put an explanation at line 37 to make it more understandable: "Therefore, if the ZHD contains bias, this error will probably be transmitted to the ZWD, which furtherly exert an influence on the total delay and the final solutions." |

| | | | |
|---|---|---|---|
| | | there. | |
| 5 | | Line 88: Put a space between "k" and "represent" | Sorry for my negligence, and we revised it. | We put a space between "k" and "represent" at Line 96. |
| 6 | | Table 1: Although you cite the climate type in the Note below the table, I don't understand why you put them in the table. If it is important to know the climate type of the Sites, please define them in a short paragraph, probably as a Note under the table. | Thanks for your advice, and we added an explanation why the climate type matters in this study. | We added a sentence at Line 134 under Table 1: "Climate type determines the general weather conditions in this area; thus, it probably keeps close relation with ZHD. Since this, the climate type of each site is displayed here for further analysis, which complies with Köppen-Geiger climate classification." |
| 7 | | Table 1: There is no info for the Climate Type and Climate name for ST07 to ST12. Probably since they are in the middle of the ocean, this information is not available. If yes, please mention it in the text; otherwise, fill in the blanks in the table with the correct info. | You're right about it, and oceans indeed don't own any climate type in Köppen-Geiger's model. We explained that under Table 1. | We added a sentence at Line 136 under Table 1: "ST07 to ST12 are located in the middle of ocean, which have no type attribution in Köppen-Geiger's classification, so a '—' was left in the table above" |
| 8 | | Line 140: Please elaborate on "half annual items in total energy". It is not clear to the reader. | Thanks for your reminder, and we changed it to another explanation. | We changed it to another sentence at Line 154: "proportion of annual and semi-annual periodic terms in total energy spectrum". |

| | | | | |
|---|---|---|---|---|
| 9 | | Line 142: There is a typo: lager -> larger | Sorry for the silly mistake, and we revised it. | We revised the word at Line 157. |
| 10 | | Line 208: Please explain in the text why you provided global distributions of $\sqrt{a_1^2+a_2^2}$ and $\sqrt{a_3^2+a_4^2}$. Is there any correlation between them? | Thanks for your suggestion, and we added a description in the manuscript. | We added a sentence at Line 232: "where $\sqrt{a_1^2+a_2^2}$ and $\sqrt{a_3^2+a_4^2}$ denote the amplifications of annual and semi-annual periodic terms, respectively". |
| 11 | | Line 212: Please replace "." with ":" after the word "conclusions". | Thanks for your reminder, and we revised it. | We revised the punctuation at Line 237. |
| 12 | | Figure 9: What is the power of 10 in the left figure, Y-axis? | Sorry that we cut too much on the left side, and we revised it. The power of 10 in the left figure is 4. | We revised Figure 9 at Line 285. |
| 13 | | Figure 9: Are the figures for "Before Corrections" and "After Corrections" overlapped? If yes, please change the style of one of them to show it better. | You're right about it. We enlarged the two figures and made the colour blocks translucent, so that they can be distinguished better. | We enlarged the two figures and made the colour blocks translucent at Line 285. |
| 14 | | Why did you consider only the SAASD for validation? What is the impact of your proposed correction model in SAASZ? | Thanks for your question. We chose $SAAS_D$ just for an example since the difference between the two models is not that large. | We added an explanation at Line 152 and added a paragraph at Line 203, which shows from another point of view that the effect of taking $SAAS_Z$ model as an example will be similar with that of $SAAS_D$. |

[Figure]

Results from ECMWF                    Observations from in-situ stations

Appendix figure 1. Surface temperatures at 2734 sites

[Figure]

Results from ECMWF                    Observations from in-situ stations

Appendix figure 2. Surface pressures at 2134 sites

[Figure]

Differences of temperature between results from          Differences of pressure between results from

ECMWF and in-situ stations                              ECMWF and in-situ stations

Appendix figure 3. Frequency distribution histogram of difference of the two datasets

**Reference**

Aminreza, N., Shahin, S., and Ahmad, S.: Evaluation of the ECMWF Precipitation Product over Various Regions of Iran, Journal of Meteorological Research, 35, 1125-1135, 10.1007/s13351-021-1093-z, 2021.

Aparecido, L. E. d. O., Rolim, G. d. S., Moraes, J. R. d. S. C. d., Torsoni, G. B., Meneses, K. C. d., and Costa, C. T. S.: Accuracy of ECMWF ERA-Interim Reanalysis and its Application in the Estimation of the Water Deficieny in Paraná, Brazil, Revista Brasileira de Meteorologia, 34, 1-14, https://doi.org/10.6084/m9.figshare.11757186.v1, 2019.

Boccara, G., Hertzog, A., Basdevant, C., and Vial, F.: Accuracy of NCEP/NCAR reanalyses and ECMWF analyses in the lower stratosphere over Antarctica in 2005, Journal of Geophysical Research: Atmospheres, 113, https://doi.org/10.1029/2008JD010116, 2008.

Decker, M., Brunke, M. A., Wang, Z., Sakaguchi, K., Zeng, X., and Bosilovich, M. G.: Evaluation of the Reanalysis Products from GSFC, NCEP, and ECMWF Using Flux Tower Observations, Journal of Climate, 25, 1916-1944, 10.1175/jcli-d-11-00004.1, 2012.

Tian, F., Li, Y., Zhao, T., Hu, H., Pappenberger, F., Jiang, Y., and Lu, H.: Evaluation of the ECMWF System 4 climate forecasts for streamflow forecasting in the Upper Hanjiang River Basin, Hydrology Research, 49, 1864-1879, 10.2166/nh.2018.176, 2018.